# Spatiotemporal Pattern and Aggregation Effects of Poplar Canker in Northeast China

He Yan [1,2], Liyuan Chen [3,4], Quansheng Ge [5], Chengming Tian [1] and Jixia Huang [1,5,*]

[1] Key Laboratory for Silviculture and Conservation of Ministry of Education, Beijing Forestry University, Beijing 100083, China; sfzzyh@126.com (H.Y.); chengmt@bjfu.edu.cn (C.T.)

[2] Key Laboratory of State Forestry Administration on Forest Pest Monitoring and Warning, Shenyang 110034, China

[3] Center for Human-Environment System Sustainability (CHESS), State Key Laboratory of Earth Surface Processes and Resource Ecology (ESPRE), Beijing Normal University, Beijing 100875, China; chenliyuan@mail.bnu.edu.cn

[4] School of Natural Resources, Faculty of Geographical Science, Beijing Normal University, Beijing 100875, China

[5] Key Laboratory of Land Surface Pattern and Simulation, Chinese Academy of Science, Beijing 100101, China; geqs@igsnrr.ac.cn

[*] Correspondence: huangjx@bjfu.edu.cn; Tel.: +86-185-1544-6101

**Abstract:** *Research Highlights:* This study looks at poplar canker caused by *Cytospora chrysosperma* as a geographical phenomenon, and it applies spatial statistics to reveal the pattern and aggregation effects of the disease on a large scale in time and space. The incidence area of poplar canker in Northeast China has spatial (spatiotemporal) aggregation effects, which emphasize the importance of coordinated prevention. The results of spatial and spatiotemporal clusters can guide specific regional prevention and indicate the possible predisposing factors, respectively. *Background and Objectives:* Poplar canker, a harmful forest biological disease that is widespread throughout Northeast China, brings enormous ecological and economic losses. The limited cognition of its spatiotemporal pattern and aggregation effects restricts the decision-making for regional prevention and the identification of disease-inducing conditions. This study aims to explore the spatiotemporal pattern and to detect the aggregation effects of the disease, trying to provide references for prevention. *Materials and Methods:* According to the incidence data of poplar canker reported by each county in Northeast China from 2002 to 2015, we mapped the distribution of the incidence rate in ArcGIS and performed retrospective scan statistics in SaTScan to detect the spatial and spatiotemporal aggregation effects of the incidence area. *Results:* The spatiotemporal pattern of poplar canker's incidence rate presents the characteristic of "outbreak-aggregation-spread-stability." The incidence area of the disease when we performed spatial aggregation scan statistics showed the primary cluster covering Liaoning province (LLR = 86469.86, $p < 0.001$). The annual spatial scan statistics detected a total of 14 primary clusters and 37 secondary clusters, indicating three phases of aggregation. The incidence area of disease also shows spatiotemporal aggregation effects with the primary cluster located around Liaoning province, appearing from 2009 to 2015 (LLR = 64182.00, $p < 0.001$). *Conclusions:* The incidence area of poplar canker presents significant characteristics of spatial and spatiotemporal aggregation, and we suggest attaching importance to the clues provided by the aggregation effects in disease prevention and identification of predisposing factors.

**Keywords:** poplar canker; Northeast China; spatiotemporal scanning statistics; spatiotemporal aggregation



## 1. Introduction

Poplar trees (*Populus* spp.) are widely distributed throughout the world and are the main tree species used for afforestation. However, they are susceptible to a variety of forest diseases. Poplar canker (*Cytospora* canker of poplar) caused by *Cytospora chrysosperma* (Pers.) Fr. (teleomorph: *Valsa sordida* Nit.) is one of the most widespread and severely damaging of them [1,2]. *Cytospora* canker can occur on at least 85 woody host plant species, including poplars and willows [3,4]. Although *Cytospora* canker of poplar also occurs on numerous other woody hosts, the disease caused on species other than *Populus* and *Salix* is generally caused by other species of *Cytospora*, i.e., not *C. chrysosperma*. Considering that the major causal agent of poplar canker is *C. chrysosperma* in Northeast China, the term "poplar canker" in this study refers to *Cytospora* canker of poplar caused by *C. chrysosperma*. Poplar canker is a powerfully infectious fungal disease [4,5], whose pathogen is a parasite that can be spread by wind, rain, and insects [6,7] and infects weak host plants from bark wounds caused by the climate, animals, or humans [1,3,8]. Once trees become infected, poplar canker leads to the decay and death of branches and the withering of leaves [3,8]. Symptoms of poplar canker include elongate, slightly sunken, and discolored areas in the bark, which often splits along the canker margin. Diseased inner-bark and the bark above the infected cambium may appear sunken and yellow, brown, reddish-brown, grey, or black, becoming watery and odorous as the tissues deteriorate. Its pathogen can spread and extend from the wounded bark into healthy tissues and even cause the death of the whole tree in severe cases [1].

Poplar canker causes significant ecological destruction and economic loss around the world. According to research, poplar canker infected 41% of black cottonwoods from the northern region of the "no-cottonwood" belt in British Columbia, Canada, around the year 2001 [9]. It caused the mass death of white poplars in the Hulla Valley, Northern Israel, in 2002 [10]. It also appeared to play a major role in the rapid mortality of trembling aspen in Southwestern Colorado, US, in 2006 [11]. Under the condition of abundant vegetation resources in China, poplar canker would generate more considerable ecological and economic losses. More than 53 species of poplars are cultivated in China and cover a total area of over 7 million hectares, accounting for 73% of poplar-planting areas in the world [12,13]. Poplar canker is a major forestry disease as well as the leading cause of poplar death in Northern China [2]. It is particularly severe in Northeast China, where mortality rates of poplar can reach more than 70% [6,14]. Researches and statistics from China indicate that the annual economic losses caused by poplar canker in the county ranged from 320 to 500 thousand yuan [15]. Four large-scale poplar canker disasters in Liaoning province from 2006 to 2013 resulted in a total disaster area of more than 87,000 hm$^2$, a death area of nearly 27,000 hm$^2$, and a direct economic loss of more than 2 billion yuan [7]. Moreover, poplar canker brought an average annual loss of forest ecological services valued at 1,028.59 million yuan [16]. To make the matter worse, poplar canker in Northeast China is becoming more aggravated and frequent [6,7].

Prediction and prevention are the fundamental and practical ways of fighting poplar canker [8], as there are no economically viable control methods suitable for this poplar disease [12]. Exploring the spatiotemporal pattern and aggregation effects of poplar canker is necessary for the formulation of macroprevention policies and understanding the prediction mechanisms. The predisposing factors that present spatial heterogeneity, such as environmental conditions, stand structure, and human management [6,8], likely allow poplar canker to occur in clusters through space and time. Currently, most studies on poplar canker in China and abroad focus on the molecular mechanism, incidence regularity, and control techniques [1,4]. However, limited studies have focused on the spatiotemporal patterns and aggregation effects of poplar canker on the regional scale from a geospatial perspective. Many studies of forest diseases and insect damage primarily focus on the pathogens or insects and miss the opportunity to explore the possible predisposing mechanisms of diseases and insect damage on a large scale of time and space through geographical analyses.

This study used a geographic information system (GIS) and retrospective scanning statistics in SaTScan to investigate the spatiotemporal pattern of poplar canker's incidence rate and aggregation

effects of poplar canker's incidence area in four provinces (i.e., Heilongjiang, Jilin, Liaoning, and Inner Mongolia) of Northeast China from 2002 to 2015. SaTScan is a spatial data analysis software that uses spatial (spatiotemporal) scan statistics to analyze local spatial heterogeneity. It can perform geographical surveillance of diseases and detect spatial or spatiotemporal disease clusters, as well as determine if these clusters are statistically significant. This study used this software for two primary reasons. First, spatial scan statistics are sensitive and effective, and they are the most suitable cluster detection method. Compared to initial scan statistics, spatial scan statistics use a variable size scanning window and deal with data from geographically inhomogeneous conditions. Thus, this detection method is the most practical [17]. Compared to the geographical analysis machine (GAM), the spatial scan statistics scanning window is continuously changing and can test the hypothesis without problems of multiple testing. Compared to global clustering tests, such as maximized excess events tests, spatial scan statistics have a higher power for detecting local clusters and their location [18,19]. Second, as a spatial analysis technology of epidemiology, SaTScan has been applied in existing research to analyze the spatial clusters of forest fragmentation in Southeastern US, spatiotemporal clusters of insects and pathogens in Pacific Northwest US [20], and geographic concentrations of perforated forests in Eastern US [21].

The purpose of this study was to (1) retrospectively analyze the spatiotemporal pattern of poplar canker's incidence rate in Northeast China from 2002 to 2015, (2) detect the spatial (spatiotemporal) aggregation effect of the incidence area of this disease on the regional scale, and (3) try to explore the possible predisposing factors in specific space and time based on the results, and to provide a reference for macroprevention. The results show that (1) the incidence rate of poplar canker presented a rising trend in general with an average annual increase rate of 27.62%. (2) Liaoning province was one of the four regions with the most severe disease from 2002 to 2015. (3) The spatiotemporal pattern of poplar canker's incidence rate presented the characteristics of "outbreak-aggregation-spread-stability." (4) The incidence area of poplar canker had spatial and spatiotemporal aggregation effects, and Liaoning province was usually in the primary clusters with the most significant probability of aggregative occurrence. Considering the aggregation effects of poplar canker, we should carry out regional coordinated prevention and control, rather than only apply prevention activities based on the incidence rate in a single geographic unit. The results of spatial clusters can reveal the potential high-risk areas and suggest prevention strategies adapted to local conditions for regional prevention. The results of spatiotemporal clusters can indicate the possible predisposing factors in specific space and time, such as afforestation activities and extreme weather.

## 2. Materials and Methods

### 2.1. Study Area

The study area spans 275 counties (county-level cities and districts) under the jurisdiction of Heilongjiang, Jilin, and Liaoning provinces and the Inner Mongolia Autonomous Region of Northeast China (97°12′, 135°05′ N; 37°24′, 53°33′ E) with an area of 199.14 million hectares (Figure 1). This area has extensive forest coverage with a wide variety of trees. It is not only the location of the largest natural forest area, the Northeast Forest Region composed of the Greater and Lesser Khingan Ranges and Changbai Mountains, but also contains the large-scale artificial afforestation project called the Three-North Shelterbelt Program. According to the China Statistical Yearbook on Environment (2018), the forest area in these northeastern provinces reaches 57.71 million hectares, accounting for 27.79% of the national forest area. In 2017, the afforestation area was 1.08 million hectares, accounting for 14% of the country. According to the Flora of China (2004), around 59 species of *Populus* and *Salix*, susceptible to poplar canker, are cultivated throughout the study area. The climate of the study area includes a temperate monsoon climate and a temperate continental climate, so the seasonal changes of temperature and humidity are distinct. The spatial heterogeneity of climate characteristics is evident in the study area. Its average annual temperature ranges from −3.6 °C to 11 °C, and the annual

precipitation ranges from 64 to 1100 mm. The climate in Northeast China from March to June is suitable for the occurrence and spread of poplar canker. This disease usually occurs in late March and April every year, and its incidence rate reaches its peak of incidence between May and June [6].

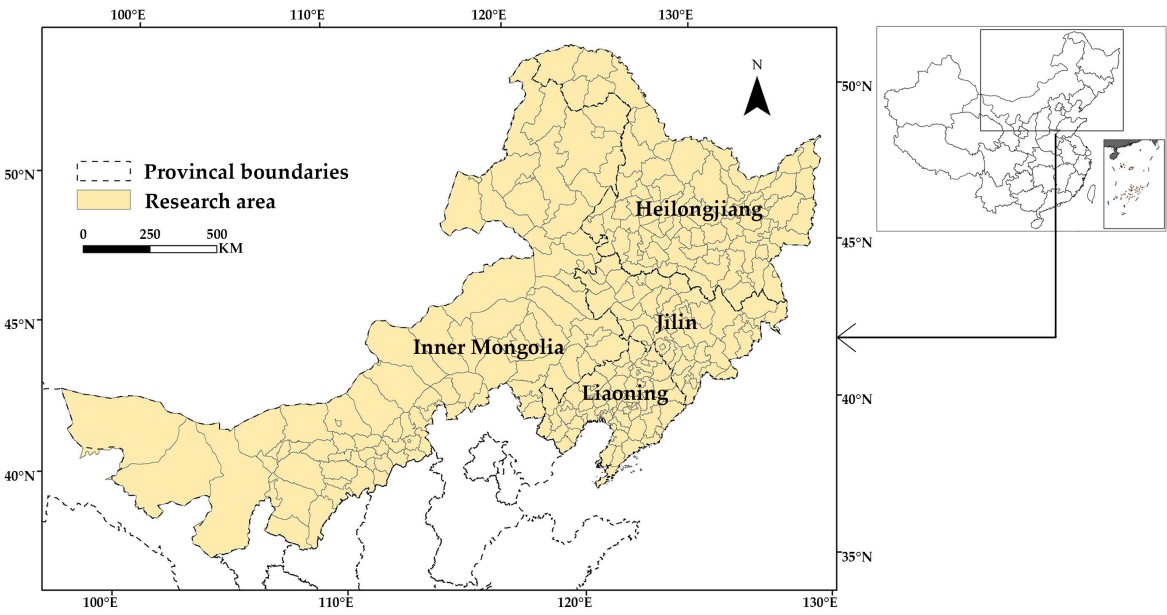

**Figure 1.** Location of the study area.

## 2.2. Data Collection and Preprocessing

The statistical data of poplar canker's incidence in the study area from 2002 to 2015 was provided by the Forestry and Grassland Diseases and Pests Control Station, National Forestry and Grassland Administration (NFGA) of the People's Republic of China. Only data collected for the occurrence of poplar canker on *Populus* species caused by *Cytospora chrysosperma* were used for the analyses. The data was acquired from the 275 detection and reporting stations set up by the NFGA at the county-level in four provinces of Northeast China. According to the main objectives of the survey and reporting methods promulgated by the NFGA, regular survey and perennial monitoring are adopted to accurately record the number of affected trees, occurrence areas, severity, and incidence rate of poplar canker every year. The types of *Populus* forests and plantings examined in the conducted surveys included plantations, non-managed planted forests, and naturally regenerated forests. The field survey was carried out using a systematic line survey and was augmented with data collected from standard circular plots (0.2 hm$^2$). The additional plots were established when a surveyed area was found to have more than 10% affected trees. Plots were set up every 5 hm$^2$ for these situations. Representative trees with poplar canker were selected for sampling and verification of the causal fungus. Specifically, isolations were made from stem cankers in the laboratory to confirm the presence of *C. chrysoperma*. The incidence rate equals the number of trees confirmed to be infected with poplar canker caused by *C. chrysosperma* divided by the total number of trees surveyed × 100%. The county-level survey data is reported to the NFGA level-by-level each year within the stipulated timeframe. With strict quality control of Forestry and Grassland Administration at four administrative levels (e.g., the county, municipal, provincial, and national levels), the accuracy rate of the data can reach more than 99%.

Additionally, the normalized difference vegetation index (NDVI) used in this study was the MODND1M monthly synthesis product with a spatial resolution of 500 m, provided by the Geospatial Data Cloud site, Computer Network Information Center, Chinese Academy of Sciences (http://www.gscloud.cn). It was initially obtained by the moderate-resolution imaging spectroradiometer (MODIS) sensor on the TERRA satellites. NDVI ranged from −1 to 1, the positive value indicates vegetation, and its value increases with the increase in the proportion of vegetation coverage.

Considering that the actual host area and total incidence area of each county have no records available, we assumed the mean value of the maximum NDVI for each month from March to July (i.e., the period when trees are luxuriant, and poplar canker is prone to occur) to approximately represent vegetation coverage, as the positive relationship between NDVI and vegetation coverage [22]. It is a typical representation in regional geographic analyses [23,24]. Under the premise that a variety of host plants (around 59 species of *Populus* and *Salix*) are widely distributed in the study area, the vegetation coverage was approximately considered as the host vegetation coverage. Thus, the host area was estimated by the product of county area and vegetation coverage, and it included all hosts of poplar canker. Then, we used the incidence rate recorded in official statistical data to multiply the host area as the incidence area. The county is the geographic unit of the calculation process. Although this process cannot accurately calculate the exact value, it reasonably represents the corresponding quantitative relationship and the actual meaning of the variables between different geographic units. The scan statistics in this study used the obtained incidence area as the variable to detect.

*2.3. Data Analysis*

2.3.1. Spatial (Spatiotemporal) Scan Statistics

This study applied scan statistics to detect clusters of cases, and it scanned the entire area by circular windows with variable positions and sizes. The windows, in turn, center on each geographical unit positioned throughout the study region, while the radius of the window varies continuously in size from zero to the upper limit, which specifies the percentage of the maximum total population at risk within the scanning window [17]. The numerous generated geographical circles are possible clusters waiting for detection. Therefore, to test the aggregation effect of each window, we propose the null hypothesis as follows. There is no aggregation effect of poplar canker in the window, and the incidence area is a random variable obeying the discrete Poisson distribution (i.e., the expected incidence area is proportional to its host area and is consistent with the characteristics of forest pests and diseases) in geographical location, independent of whether it is inside or outside the window. The corresponding hypothesis is that the incidence area in the window is abnormally higher than outside the window. The log-likelihood rate (LLR) is a statistic that describes the degree of abnormality of the incidence in the window. Under the condition of a discrete Poisson model, the likelihood function for a specific window is proportional to [17]:

$$(a/E(a))^a \times ((A-a)/A-E(a))^{A-a} \times I(), \tag{1}$$

where A is the total area of poplar canker incidence, a is the observed area of incidence within the window, and E(a) is the covariate-adjusted expected area of incidence within the window under the null hypothesis. I () is an indicator function that is equal to 1 when the window has areas that are larger than expected under the null hypothesis, and 0 otherwise.

The cluster with the largest calculated LLR (i.e., the cluster least likely to be an accidental infection of poplar canker) is the primary cluster (i.e., the most likely cluster), of which the likelihood ratio is used as the maximum likelihood ratio test. The remaining statistically significant scan windows are secondary clusters, sorted by LLR in descending order. Monte Carlo simulation was used to generate random datasets under the null hypothesis. The *p*-value was calculated by comparing the rank of the maximum likelihood from the realistic dataset with the maximum likelihoods from the random datasets [17]:

$$p = R/(1+N), \tag{2}$$

where R is the rank of the realistic dataset and N is the number of random datasets generated by Monte Carlo simulations. When the *p*-value rejects the null hypothesis, poplar canker in the scanning window can be considered to have significant aggregation effects, and the relative risk (RR) is meaningful.

RR represents how much more severe the disease is in this location and time compared to the baseline. The higher the RR value, the more seriously poplar canker aggregately occurred:

$$RR = (a / E(a)) / ((A\text{-}a / A\text{-}E(a)), \tag{3}$$

The spatiotemporal scanning statistics add a time dimension based on the spatial scanning statistics and replace circular windows with cylindrical windows. Their bottom had the same meaning and function as the circles of the spatial scan statistics, while the height reflects the period of the possible cluster. The position, radius of the bottom, and height of the cylinder windows are variable and cover the entire space and study period.

### 2.3.2. Scan Statistics Using SaTScan

SaTScan v.9.5 was developed by Martin Kulldorff at Harvard University Medical School in Boston, MA, U.S., together with Information Management Services Inc. (www.satscan.org). It was the tool used for the spatial (spatiotemporal) scan statistics. The three input files need to contain the spatial coordinates of a set of geographic units, the number of cases in each unit, and the population size for each unit, respectively. They link to each other through a standard county code, which also served as the unit identification. Similar to the existing application of SaTScan in forest health research [20,21], this study used the poplar canker's incidence area as the "case" and the host area as the "population".

A pre-experiment was used to identify the maximum radius of the scanning window suitable for this research. It found many non-pathogenic areas located in the cluster and potentially meaningless secondary clusters could both be avoided when the window size was set to 30% of the total host area (compared to 40% and 50%). This conclusion is consistent with a previous study [25]. Therefore, the upper limit of the maximum radius in the scan statistics was set to 30%.

The number of Monte Carlo simulations was set to 999 so that the calculated *p*-value would indicate whether to reject the null hypothesis when compared to the typical cut-off value of 0.001. Furthermore, we kept the other default settings so that geographically overlapping clusters were not reported.

The SaTScan output results include result report documents and geospatial information. All statistically significant clusters, including the geographic units, aggregation center, radius, LLR, RR, and other statistical variables, were recorded in the documents. The generated ".shp" files were used for the ArcGIS mapping. In the map, a circle represents a spatial cluster unless the cluster is confined to only one county, in which case it is represented by a dot.

## 3. Results

### 3.1. The Incidence Trends of Poplar Canker

The average incidence rate (AIR) of poplar canker in Northeast China exhibited a fluctuating upward trend from 2002 to 2015 (Figure 2a). AIR sharply increased with an average annual increase rate of 27.62%. The disease was rapidly aggravated in 2003 and 2006 when the AIR increased by 1404% and 164% compared to the former year, respectively. The AIR increased steadily over five years from 2009 to 2014 with an average annual increase rate of 14.87%, reaching a maximum value of 4.54% during the study period in 2014.

During the study period, a total of 901 poplar canker cases in 149 counties was reported in the four provinces of Northeast China. The fluctuation trend of the number of infected counties was highly consistent with the AIR trends, except for 2006 (Figure 2b). The AIR of infected counties in most years was between 6.63% and 14.86% but was 31.74% in 26 infected counties in 2006. Similarly, the number of infected counties rose to its maximum observed value during the study period in 2014 when 100 counties were affected.

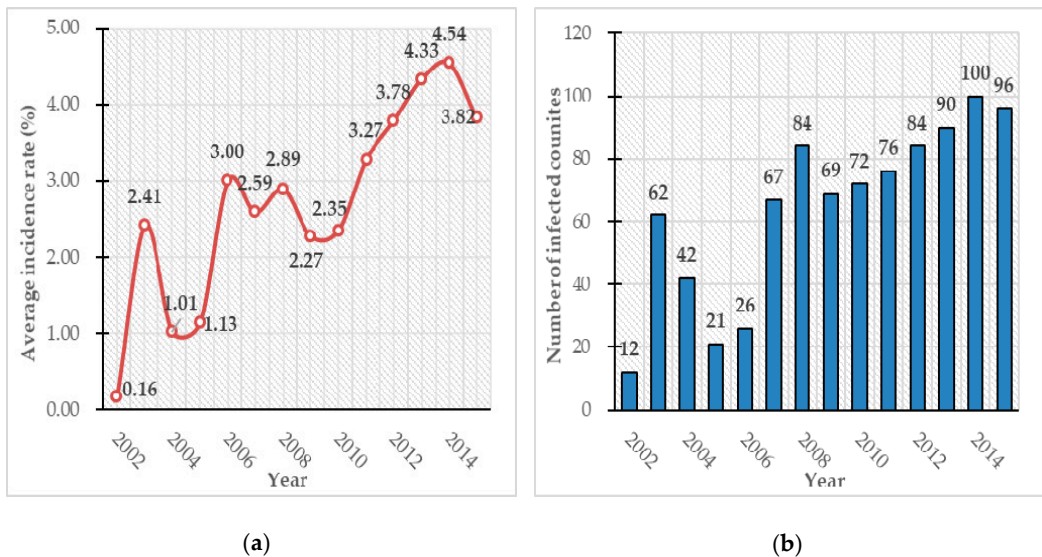

(**a**)                                                    (**b**)

**Figure 2.** (**a**) Annual variation of the average incidence rates of all counties in the study area from 2002 to 2015; (**b**) Annual variation of the number of infected counties in the study area from 2002 to 2015.

### 3.2. The Severity of Poplar Canker in Affected Counties

High incidence rates of poplar canker were mainly concentrated in four regions as follow: (1) Liaoning province, (2) western counties in Jilin province, (3) Ulanqab, Baotou, and their surrounding areas (region U-B), and (4) Jarud and Ar Horqin banners (region J-A) (Figure 3). Some others were distributed in southern Heilongjiang and southern Jilin.

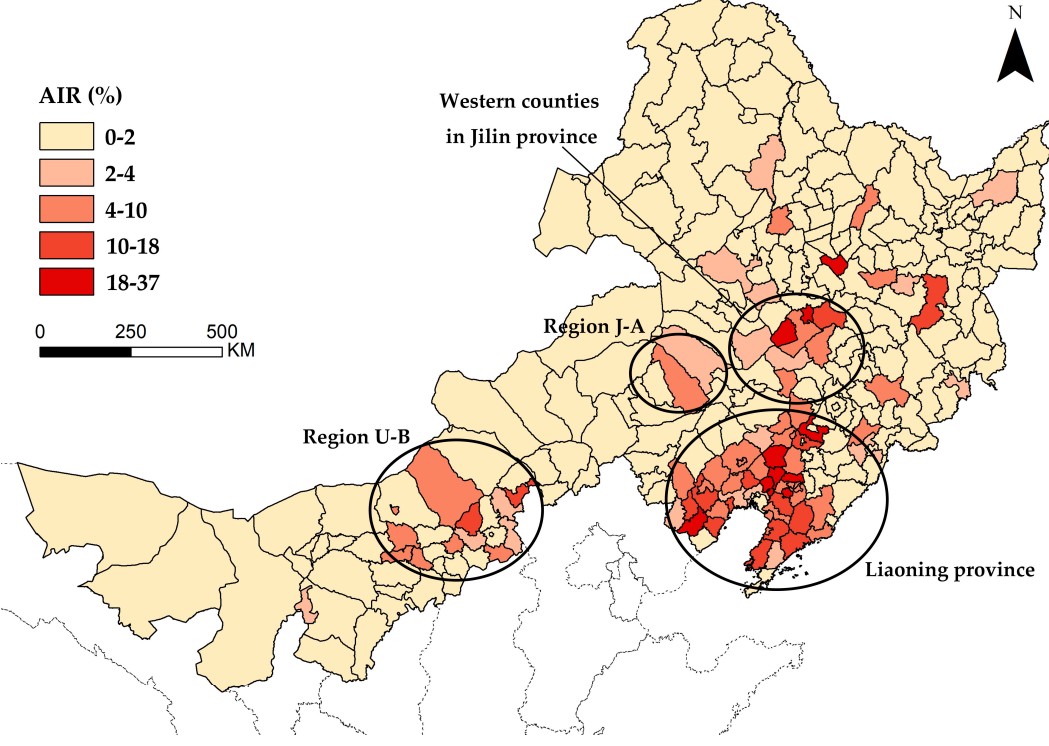

**Figure 3.** The average annual incidence rates of poplar canker in each county of Northeast China, reflecting the overall severity of the disease during the study period.

Poplar canker in central Liaoning province had a high incidence rate, and it continually occurred for a long time in this area. The AIR in Xinmin city, Liaozhong county, Taian county, Dengta city, and the Anshan municipal district were all over 18%. Liaozhong county and three other counties suffered from the disease for 13 consecutive years. Another seven counties, including Xinmin city, suffered from the disease for 12 years during the study period. The AIR of five counties in Western Jilin, led by Songyuan municipal district and Qian'an county, ranged from 7% to 25% and only occurred from 2007 to 2008 and 2012 to 2015. In region U-B, the AIR ranged from 2% to 12.8% in many areas, and the first case of poplar canker was reported in 2010. Then, the disease spread and became exacerbating around 2012. In region J-A, poplar canker broke out at the beginning of the study period and continued for a more extended period compared to region U-B, and its AIR ranged from 2.2% to 5.4%.

### 3.3. The Spatiotemporal Pattern of Poplar Canker

The spatial distribution pattern of poplar canker's incidence rate in Northeast China presented "outbreak-aggregation-spread-stability" (Figure 4). The "outbreak" stage began in 2002 when Chifeng city, Tongliao city, and a few of their neighboring eastern counties reported their first poplar canker cases. This phase lasted until 2003 when poplar canker was not only introduced to 37 counties in Liaoning and Jilin provinces but had a wide dissemination range and even spread to 16 counties located in Northern Heilongjiang and Western Inner Mongolia, rendering all provinces in Northeast China infected by poplar canker.

Over the next three years, notable contractions occurred in the affected areas, and the disease entered the "aggregation" stage. Particularly in 2006, infected counties, such as Nong'an county, Liaozhong county, Anshan municipal district, and Haicheng city, were centrally located in central and northern Liaoning province and Western Jilin provinces with an extremely high incidence rate over 70%.

The "spread" phase lasted from 2007 to 2014. The spatial extent and incidence rate of the poplar canker repeatedly increased and decreased before 2011. Then, the disease aggravated steadily after 2011, some counties at the junction of Songyuan city and Daqing city were infected, and their incidence rates were high. The same situation poplar canker also occurred in the U-B region. From 2014 to 2015, the condition of poplar canker was relatively stable, and the state of the most infected counties remained approximately unchanged except for a few counties located in Eastern Inner Mongolia.

### 3.4. Detection of Spatial Aggregation Effects

The spatial scan of the cumulative incidence area of poplar canker identified one primary cluster and eight secondary clusters (Figure 5). The primary cluster was located in Liaoning province and its western contiguous zone with a radius of 234.88 km was geographically centered around Beizhen city ($RR = 9.68$, $p < 0.001$). The secondary cluster 1 was located in the junction of Western Jilin and Southwestern Heilongjiang with a radius of 136.41 km centered around Zhaoyuan county ($RR = 4.63$, $p < 0.001$). Secondary clusters 2 and 3 were each located in one county, Hailin city ($RR = 8.07$, $p < 0.001$) and Fuyu county ($RR = 5.93$, $p < 0.001$), respectively. Secondary cluster 4 covered region U-B and four counties of Xilin Gol league and Hohhot city ($RR = 1.98$, $p < 0.001$), centering on Siziwang banner with a radius of 213.59 km. The remaining secondary clusters were scattered in Jarud banner and Huolin Gol city and other places with relatively low incidence rates but significant aggregation effects ($RRs = 1.49$–$3.32$, $p < 0.001$).

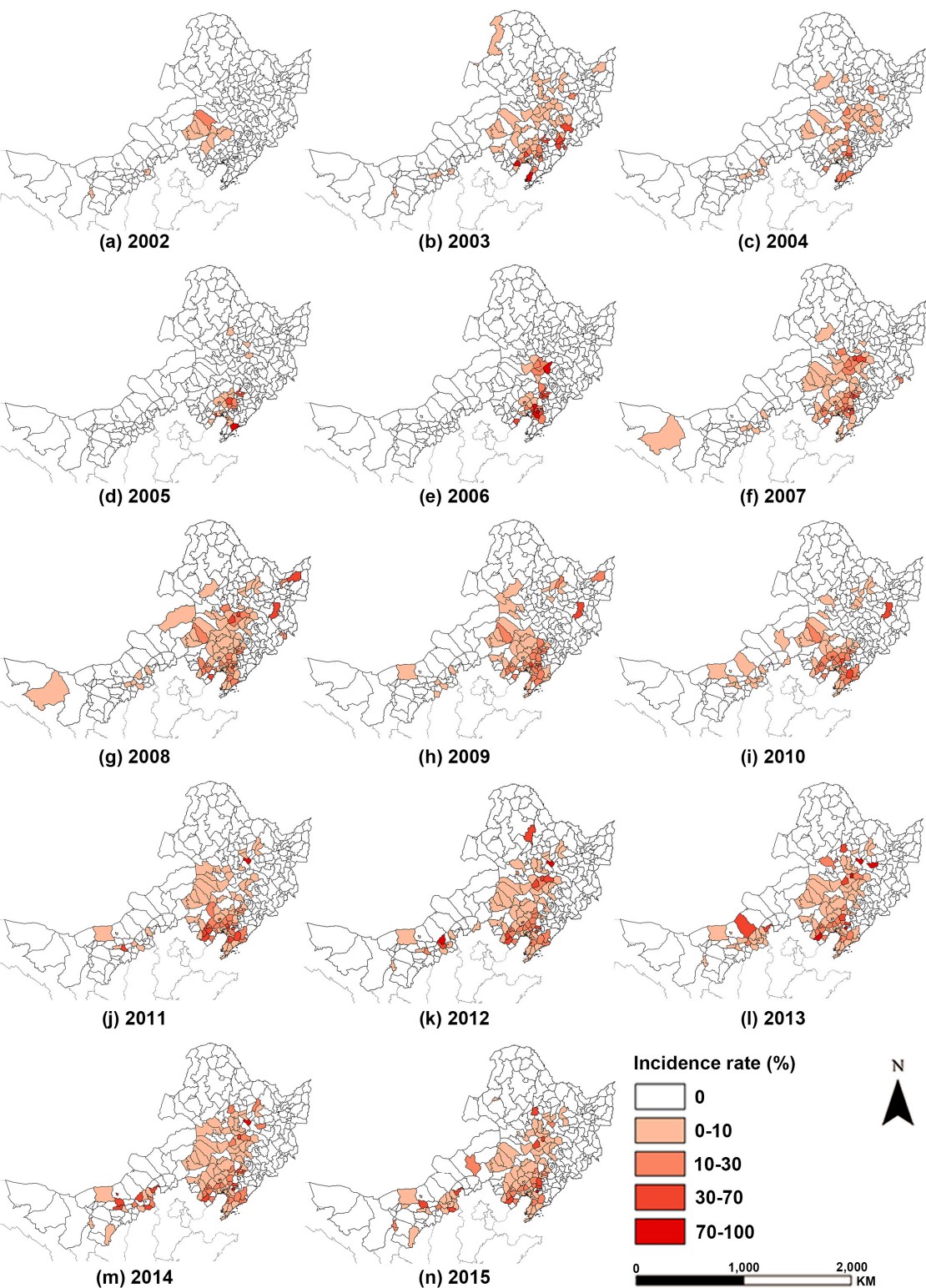

**Figure 4.** (**a**) to (**n**) correspond to annual variation of the spatial distribution of poplar canker incidence from 2002 to 2015 respectively.

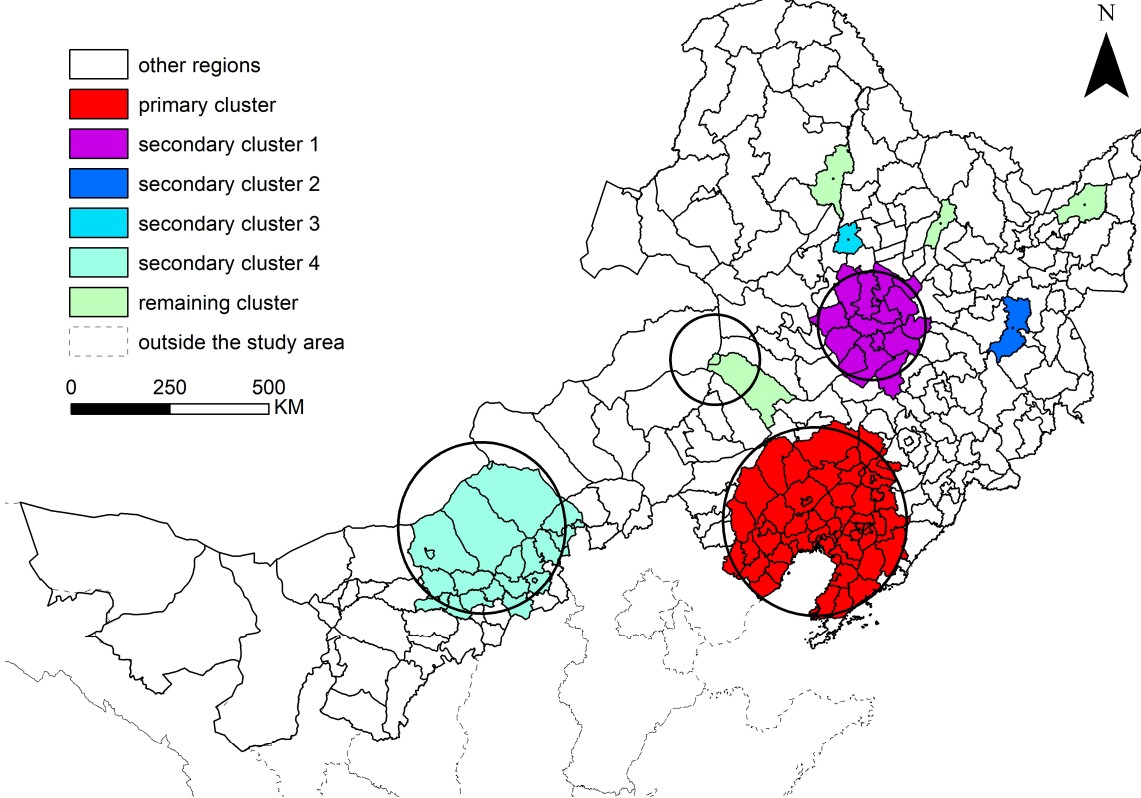

**Figure 5.** The significant clusters of cumulative poplar canker incidence are identified by spatial scan statistics.

Purely spatial scan statistics conducted for each year detected 14 primary clusters and 37 secondary clusters (Appendix A Table A1), indicating three phases of aggregation (Figure 6). The first stage occurred from 2002 to 2007, the number of detected clusters was small, and their location was singular during this period. The primary cluster presented regular changes in the aggregation radius and number of counties exhibiting the "expansion-centralization-expansion" pattern with the movement of the aggregated barycenter. The aggregation area of disease extended southeastward to Liaoning province and Jilin province. After the disease concentrated in Liaoning province, its aggregation effects expanded again. There were only a few irregular secondary clusters, and these clusters did not represent the risk of further aggregation at the time. The second stage occurred from 2008 to 2009, when Hailin city was severely infected and replaced the Liaoning province and Jilin province clusters as the primary cluster. The latter became secondary cluster 1 and exhibited a decreasing trend of the radius, thereby moving the barycenter to Liaoning province. More secondary clusters appeared in Heilongjiang province during this period. The third stage occurred from 2010 to 2015, the location and coverage of the primary cluster each year were relatively stable, and the number of secondary clusters increased. New stable clusters formed around Songyuan city and Ulanqab city, respectively.

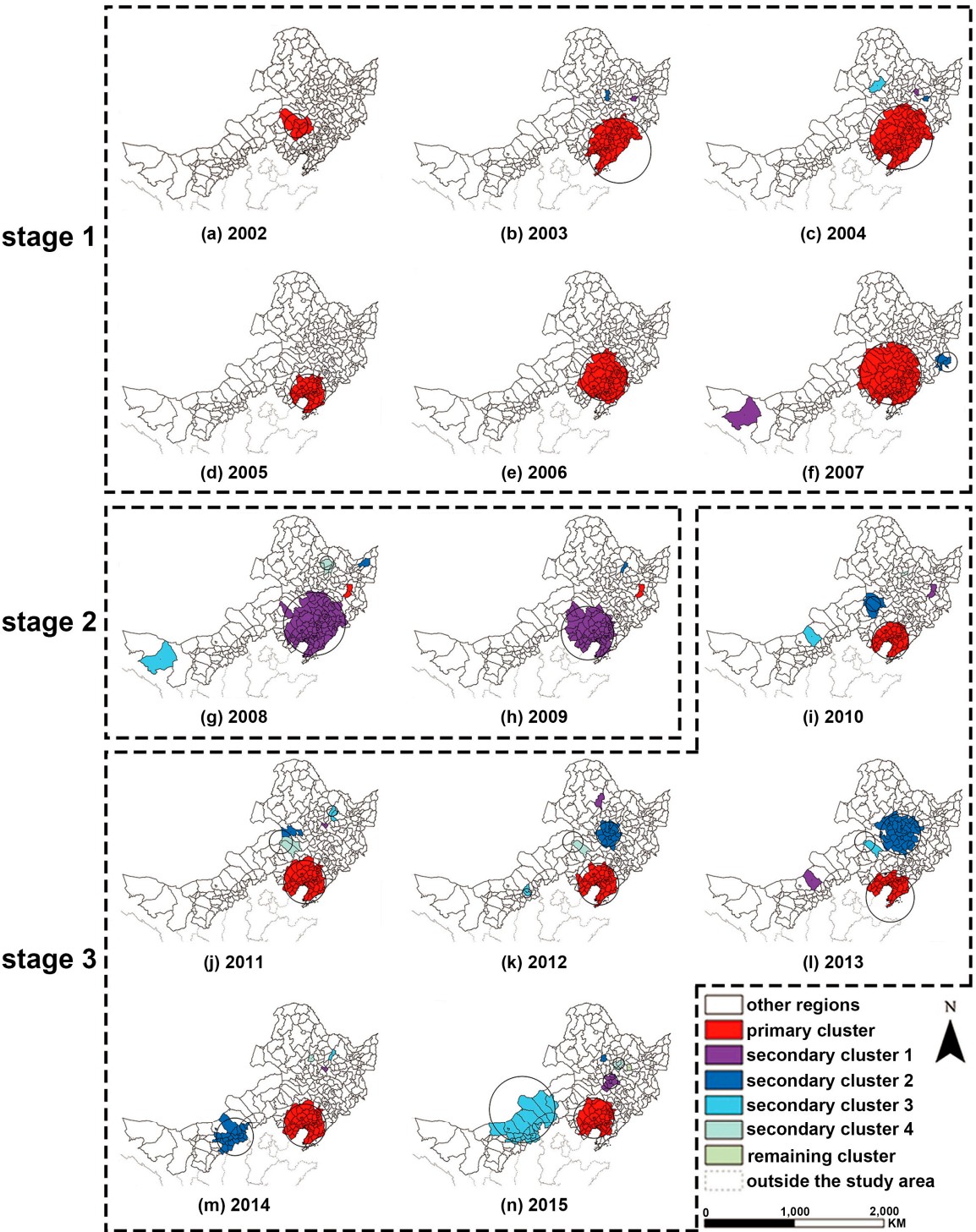

**Figure 6.** (**a**) to (**n**) correspond to annual variation of spatial distribution clusters detected by spatial
scan statistics from 2002 to 2015 respectively.

*3.5. Detection of Spatiotemporal Aggregation Effects*

The incidence area of poplar canker in Northeast China also had significant spatiotemporal
aggregation effects. Spatiotemporal scan statistics detected one primary cluster and six secondary
clusters (Figure 7). The primary cluster covered most counties throughout Liaoning province and
its surrounding counties of Southeastern Tongliao city, occurring from 2009 to 2015, with a radius of
241.89 km and a center in Beizhen city. The poplar canker incidence of this cluster was relatively higher

than the outside (*RR* = 9.51, *p* < 0.001). Secondary clusters 1 and 2 contained only Hailin city from 2008 to 2010 and Lanxi county from 2011 to 2015, respectively. Both of them had an extremely high risk of aggregated disease (*RRs* = 38 and 56.95, respectively, *p* < 0.001). Secondary cluster 3 covered Songyuan city in Western Jilin province and its neighboring Nong'an county, Da'an city, and Zhaoyuan county, where the concentration of the poplar canker occurred from 2006 to 2008, had a radius of 90.36 km, and a center in Qian Gorlos county. This cluster also had a very high incidence risk (*RR* = 14.36, *p* < 0.001). Secondary cluster 4 was located in region U-B and its neighboring counties of Hohhot city and Xilingol league from 2012 to 2015, centering on Siziwang banner with a radius of 213.59 km. Its incidence risk was relatively low (*RR* = 5.74, *p* < 0.001). The remaining clusters were mainly concentrated in region J-A in 2008 with a low risk of aggregated disease (*RR* = 4.53, *p* < 0.001) and scattered throughout the Mo banner in 2012 with high severity (*RR* = 32.86, *p* < 0.001).

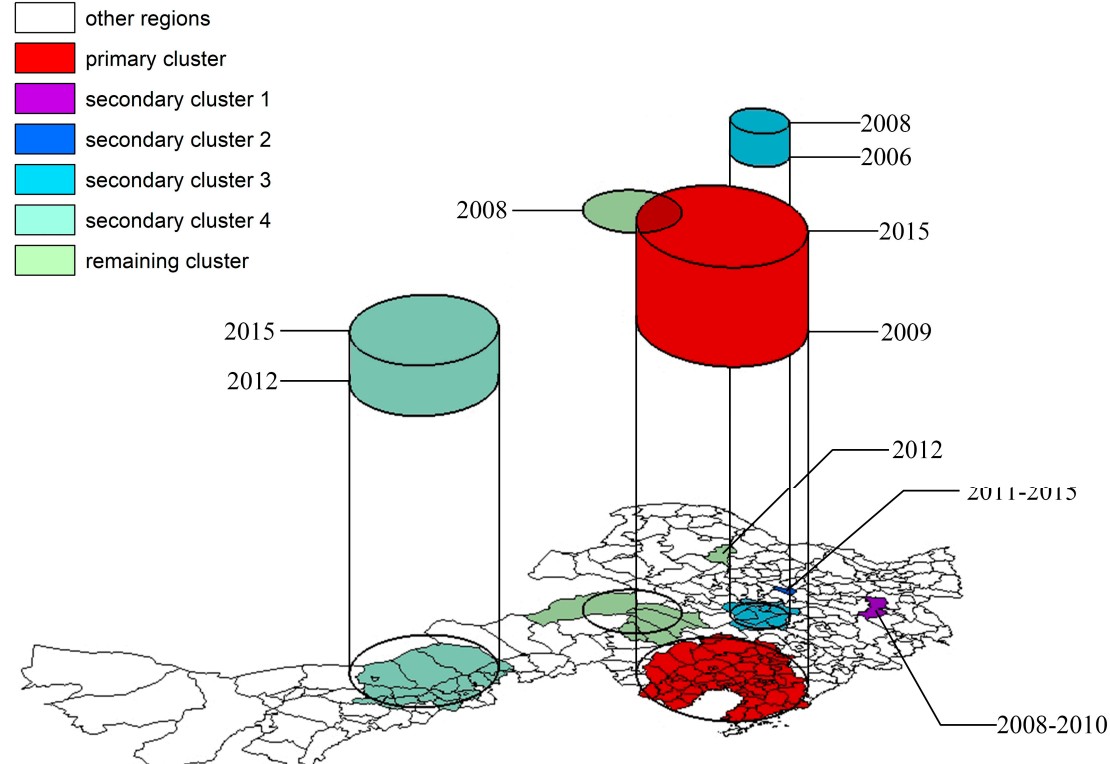

**Figure 7.** The location and time intervals of clusters detected by spatiotemporal scan statistics.

## 4. Discussion

The extent and severity of poplar canker in Northeast China exhibited an upward trend during the study period. Partial reasons for this phenomenon are the natural spread of disease and our poor performance in prevention and control. Apart from this, we need to pay attention to a series of afforestation projects in China, including the Three-North Shelterbelt, Beijing–Tianjin Sand Source Control, and Grain to Green, which can exacerbate the spread of the disease. The young seedlings of poplars are highly likely to carry pathogens [14] and are weak and susceptible at the sapling stage [1]. It is easy to introduce a new source of poplar canker during cultivation. Currently, people tend to focus on the greening benefits of these forestry projects [26] but ignore the survival rate of planted trees and the forest diseases and pests that may occur when trees are transplanted or imported under intensive and homogeneous afforestation [8,14]. Research has found that forest diseases like poplar canker are the leading cause of degradation in the Three-North Shelterbelt forests of Liaoning province [27].

As far as we know, previous studies have not reported the spatiotemporal pattern as "outbreak-aggregation-spread-stability" for poplar canker's incidence rate in Northeast China. This pattern may be a region-specific pattern as predisposing factors with the characteristics of spatial

heterogeneity, such as daily average temperature and humidity, abnormal climate, soil characteristics, and human management, can induce poplar canker [6]. Thus, it is inadequate to explore the possible predisposing factors without considering the spatial heterogeneity of incidence. Several studies on occurrence and factors of forest disease have focused on the spatial heterogeneity [28–31]. This study further detected spatial heterogeneity through spatial and spatiotemporal scan statistics.

This study identified the spatial and spatiotemporal aggregation effects of poplar canker's incidence area in Northeast China for the first time. Moreover, the spatial aggregation effects have existed since the initial report of poplar canker. Our findings support the previous conclusion that many poplar pathogen outbreaks have occurred at the regional scale [13]. The counties with a high possibility of aggregation effect of poplar canker detected by spatial scan statistics included not only counties with high incidence but also some surrounding geographical units with relatively low incidence. This suggests that we should neither merely regard the incidence rate in the unit as the standard for prevention nor treat poplar canker independently in each unit. Instead, we need coordinated prevention on the regional scale using the reference of aggregation clusters. On the one hand, the incidence rate may not accurately reflect the actual condition of poplar canker, considering its characteristic of latent infection [11]. On the other hand, the disease can quickly spread to neighboring areas by natural conditions and human activities. The participation of multiple counties is more conducive to systematic and effective prevention.

The knowledge extracted from the interannual change pattern of spatial aggregation clusters can guide regional prevention strategies. Aggregated poplar canker occurred year-round in Liaoning province, which had the highest incidence risk and should receive continued attention, as well as be given priority in prevention and control work. The clusters gradually diversified, and new clusters separated from Western Jilin province. In addition, the region U-B also progressively developed into new clusters. Although the severity of poplar canker in these clusters was relatively weak during the study period, there were high potential risks of further expansion. Thus, prevention work in these clusters and their surrounding areas should be strengthened. The clusters detected in only a single county were usually accompanied by high incidence and short duration, mostly caused by specific acute disturbance. Emergency disposal and improving the effectiveness of prevention should be the focus in these regions.

Different from the panel data used in the purely spatial scan statistics, spatiotemporal scan statistics detect the aggregation effect in the continuous temporal dimension. The latter is more conducive to identifying the possible predisposing factors in specific space and time, in particular, factors such as climate and policy with long duration and delayed response. For example, poplar canker in Liaoning province exhibited spatiotemporal aggregation effects from 2009 to 2015. The probable explanation is that this is due to afforestation and extreme weather in Liaoning province during that period. At that time, Liaoning province planted *Populus × canadensis* No.107 and No.108, which are vulnerable to freezing injury and cannot adapt well to the climate conditions in Liaoning province [7]. Moreover, Liaoning province frequently suffered from disasters of drought, flood, and other abnormal climatic conditions, including cold spells in late spring during these years (China Meteorological Disaster Yearbook, China Meteorological Administration). The above conditions can wound trees and obstruct their normal physiological processes [6,8], rendering it easy for canker to infect poplars and form into the aggregated clusters.

There are some limitations to this study. Firstly, the scan statistics fixed the shape of the window as a circle, which may result in a bias of the location of clusters, because it is insensitive to geographic units separated by linear features, such as mountains or rivers. Subsequently, the grain of time in our data is slightly coarse, and it is difficult to reflect the seasonal changes of poplar canker and corresponding factors. Additionally, this study did not quantitatively explore the precise impact of influential factors on incidence, which can be studied by geographically weighted regression (GWR) in future research. Furthermore, accurate prediction is the fundamental goal of forest disease prevention

and control. The prospective spatiotemporal scan statistics can warn of the possibility of aggregation effects within the period, which is worth studying and utilizing in future research.

## 5. Conclusions

This study retrospectively summarized the spatiotemporal pattern of poplar canker and detected its aggregation effects. The results show that both the severity and influential area of poplar canker increased in Northeast China from 2002 to 2015. The spatiotemporal pattern of poplar canker in Northeast China presented the characteristic of "outbreak-aggregation-spread-stability." The incidence area of poplar canker had spatial and spatiotemporal aggregation effects. We suggest that coordinated prevention in a regional area is more effective, and suggest incorporating the results of spatial and spatiotemporal clusters when designating local prevention strategies and identifying the possible predisposing factors in space and time, respectively. The further application of prospective spatiotemporal scan statistics for the prediction of forest diseases and pests is expected.

**Author Contributions:** Conceptualization, J.H. and C.T.; Data curation, H.Y. and L.C.; Formal analysis, H.Y.; Funding acquisition, C.T.; Investigation, H.Y.; Methodology, L.C.; Project administration, C.T. and Q.G.; Resources, H.Y. and C.T.; Software, L.C.; Supervision, C.T. and Q.G.; Validation, H.Y., L.C., and J.H.; Visualization, L.C.; Writing—original draft, H.Y. and L.C.; Writing—review and editing, J.H. and C.T. All authors have read and agreed to the published version of the manuscript.

**Funding:** This research was funded by National Key Research and Development Program of China, grant numbers 2017YFD0600105, 2018YFC1200400, and 2019YFA0606600 and the National Science and Technology Major Project, grant number No.21-Y30B02-9001-19/22.

**Conflicts of Interest:** The authors declare no conflict of interest.

## Appendix A

**Table A1.** Summary of the purely spatial scan conducted for each year.

| Year | Primary Cluster | | | | Secondary Clusters | |
|------|----------------|---|---|---|-------------------|---|
| | Central County | Radius of Cluster (km) | Number of Counties | RR | Number of Clusters | Number of Counties |
| 2002 | Kailu | 132.11 | 8 | 219.99 | 0 | 0 |
| 2003 | Kuandian | 355.53 | 73 | 46.11 | 2 | 2 |
| 2004 | Tieling | 349.56 | 93 | 20.32 | 3 | 3 |
| 2005 | Taian | 196.36 | 45 | 4196.72 | 0 | 0 |
| 2006 | Kangping | 262.40 | 67 | 134.75 | 0 | 0 |
| 2007 | Horqin Left Wing Rear Banner | 342.10 | 95 | 33.65 | 2 | 5 |
| 2008 | Hailin | 0 | 1 | 32.92 | 4 | 101 |
| 2009 | Hailin | 0 | 1 | 45.02 | 2 | 63 |
| 2010 | Dawa | 204.48 | 46 | 16.75 | 4 | 6 |
| 2011 | Panshan | 236.85 | 55 | 25.92 | 4 | 7 |
| 2012 | Dawa | 228.86 | 50 | 8.98 | 4 | 20 |
| 2013 | Wafangdian | 271.67 | 40 | 7.90 | 3 | 40 |
| 2014 | Pansan | 236.85 | 54 | 8.79 | 4 | 24 |
| 2015 | Fuxin | 223.62 | 48 | 12.41 | 5 | 40 |

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
