# Peer review of "Spatiotemporal Pattern and Aggregation Effects of Poplar Canker in Northeast China"

_forests, doi:10.3390/f11040454_

Round 1

Reviewer 1 Report

Reviewer’s Comments

Title: Spatiotemporal transmission and aggregation effects of poplar canker in Northeast China

Overview:

The authors present results of a regional study designed to study changes in spatial distribution and temporal occurrence of poplar canker over a large geographic region (275 counties) in Northeast China from 2002 through 2015. Spatiotemporal modelling (spatial scan statistic using SaTScan software tool) of disease incidence of poplar canker was performed with data collected and provided by each county to the Forestry and Grassland Diseases and Pests Control Station, National Forestry and Grassland Administration. Overall, the authors report an upward trend in disease incidence over the time period. Expansion and contraction of the spatial extent and canker incidence rate were found in the years before 2011. Steady increase (“aggregation”) was detected between 2011 and 2014, after which the incidence and spatial extent of trees with poplar canker stabilized. The authors seem quite adept and skilled in the use of spatial and temporal analysis tools. Results of their analyses are clearly displayed through multiple figures with appropriate use of colors. Several key concerns of this reviewer are three-fold: 1) The data the authors acquired and used for their intricate analyses are very poorly described, 2) The use of several terms (e.g. “transmission”, “pathogenic factors”) are not in agreement with the standard definition of the same terms in the discipline of plant pathology, 3) The use of these terms in the Results and in interpretation of their application to prediction and management (Discussion) are questionable due to the previously listed concern, and 4) preparation of the manuscripts appears to not have involved either a well-trained, classical plant pathologist or a native English writer who has some understanding of plant pathological terms. The manuscript requires major revision, additional information (especially on how surveys were conducted, data collected, basis for calculating disease incidence), and serious re-thinking of terms used to describe the results and to interpret the results for prediction, prevention and management of poplar canker on a regional scale. Some elaboration and examples of the above concerns follow.

RE: the pathogen and the disease

The authors use “poplar canker” as the common name of the disease described as Cytospora canker of poplar (Sinclair and Lyons, 2005) caused by Cytospora chrysosperma the commonly encountered form (anamorph) of Valsa sordida on host plants. However, cankers and dieback of Populus and Salix are caused by Leucostoma niveum (syn. Valsa nivea) whose anamorph Cytospora nivea can also cause similar symptoms on the same hosts. In general, both pathogens are considered opportunistic pathogens that can cause disease on stressed on weakened hosts. However, V. sordida can more aggressive than L. niveum and, at times, severely affect vigorously growing hosts. Do the authors know if the incidence data they are using is solely for the cytospora canker fungus? Furthermore, the ability of both V. sordida and L. niveum to quickly colonize dead bark of their hosts can mask more serious diseases, such as cankers caused by Septoria musiva. Were other diseases of poplars noted during the surveys?

RE: Data collection

The authors clearly state that the data they used for their analyses was collected and summarized (as “disease incidence rate”?) by others. The data for each county were submitted to or obtained by the Forestry and Grassland Diseases and Pests Control Station, National Forestry and Grasslands Administration. That is the extent of the information given on the data that is the basis of the entire study. This poorly described portion of the Materials and Methods needs to be addressed. Was a standardized survey protocol used to obtain data by each of the 275 counties? Was that survey limited to Populus species? What percentage of the data were from plantations? How were the affected areas (e.g. plantations) surveyed? For example, were random plots established to represent a certain proportion of the area with poplars? Or, was a systematic line survey used? How was pertinent data on the disease and pathogen collected? For example was the presence of Cystospora chrysosperma fruiting bodies noted when a canker or stem dieback was recorded for a tree? How was “disease incidence rate” calculated? On a stand basis (e.g. number of affected trees/total number surveyed x 100)? Or, was it determined by presence of poplar cankers on trees on a certain area of land (e.g. a hectare)?

The authors extrapolated the disease incidence data to cover larger spatial areas (i.e. county level). Was NDVI used to distinguish Populus vegetation from other species of vegetation? Or, if any known hosts of poplar canker were present, were they included in the estimate of host area?

RE: Use of plant pathology terms

Transmission with a fungal disease such as this one is defined by plant pathologists as “the transfer or spread of a pathogen from plant to plant.” The authors did not study poplar canker in this manner. It appears that the change in disease distribution (spatial) and incidence (occurrence of poplar canker on trees or area of land?) for 2002 through 2015 was analyzed rather that “spatiotemporal transmission pattern.”

“Pathogenic factor” is not a term generally used by plant pathologist, and not used in the manner it was in this manuscript. It appears that part of what the authors are referring to is “pre-disposing factors” where predisposition is defined as “the tendency of non-genetic factors (living and non-living) acting before infection to increase the susceptibility of a plant to disease. This commonly results from pathogenic or environmental stress.” Human-related factors that might be related to whether poplar canker occurs or not might include an action such as using planting stock already infected with the pathogen during reforestation or afforestation.

Reviewer 2 Report

Even if I am a plant pathologist and not a geo-referating expert, this research sounds to me quite innovative and with commendable goals.

In the attached file, your pdf manuscript, I have put a few comments you can read by click on the highlighted words. More generally, in my opinion a proper description of the whole symptoms induce by Cytospora stem canker is lacking, since it would be useful for a better understanding of the discriminating parameters adopted for assessing disease distribution.

For the rest, I think that the paper is complete and properly discussed

Round 2

Reviewer 1 Report

REVIEWER COMMENTS

Manuscript ID: 750431

Following are comments and suggestions on the revised manuscript “Spatiotemporal pattern and aggregation effects of poplar canker in Northeast China” provided by the authors. A general overview of this reviewer’s comments were provided in a review of the original version of the manuscript and, thus, is not repeated in this review. Comments on further changes needed for version 2 of the manuscript are presented below.

  1. Comments on disease name and causal agent

The authors responded to wording and sentence changes needed regarding the disease name and causal agent. Some additional, minor changes are suggested.

a. Omit “stem” from use of the disease name “Cytospora stem canker of poplar (e.g. line 50 and line 52). Reason for change: Cytospora chrysosperma causes branch cankers as well as stem cankers on Populus species.

b. Authors should include a sentence stating the fact that although poplar canker, as it is defined in China and in this manuscript, occurs on numerous other woody hosts, but the disease caused on species other than Populus and Salix are generally caused by other species of Cytospora, i.e. NOT C. chrysosperma.

2. Further, minor clarifications are needed to describe the dataset used for the study, i.e. in 2.2 Data collection and pre-processing section.

a. Please include a sentence that only data collected for occurrence of poplar canker on Populus species caused by Cytospora chrysosperma were used for the analyses.

b. Please include a sentence on what types of Populus forests and plantings were examined in the conducted surveys that generated the data. For example, these MAY have included: plantations, shelterbelts, non-managed planted forests, naturally regenerated forests, urban forests, etc.

3. Comments on authors’ decision to globally change the term “pathogenic factors” with “causal agent”

This reviewer believes the replacement term chosen for use (“causal agent”) throughout the manuscript is incorrect for what the researchers were actually investigating. Lines 83 – 85 currently state: “Because causal agents [underlined for reviewer’s emphasis] which present spatial heterogeneity, such as environmental conditions, stand structure, and human management likely allows poplar canker to occur in clusters through space and time.” This reviewer thinks the authors are trying to show that environmental conditions (like drought stress?) are pre-disposing Populus species to infection by C. chrysosperma and subsequent canker development on portions of the geographical area covered in this study. Stand structure, e.g. high density of Populus stems, may be conducive to local spread of the pathogen (e.g. via rain-splashed and wind-driven rain dispersal of pathogen spores) leading to clustered occurrence of the disease. Human management factors that may contribute to clustering of the disease on the landscape and could include activities like pruning of branches and planting of diseased stock during reforestation or afforestation efforts. The reviewer’s point is that environmental conditions or stand structure or human management are not the causal agent in their study. Cytospora chrysosperma is the causal agent and these other factors exacerbate the disease occurrence and may be related to pattern of disease on the landscape.

4. Much editing is needed in several portions of the manuscript to address the problem with improper or un-clear use of the English language.

The reviewer started to provide suggested revisions throughout the manuscript, but the journal editor could not allow additional time for this activity. Thus, only two examples are provided to illustrate the problem and to offer a suggested revision.

Example 1  

Lines 88-91 – “So many quantities of researches about forest diseases and insect pests ignore the opportunity to serve it as a geographical phenomenon that they can't reveal the possible causal agents in specific space and time.”

This reviewer thinks the authors may be trying to say: “Many researchers investigating forest disease and insect damage patterns on the landscape are missing the opportunity to explore relationships of pre-disposing environmental conditions, stand characteristics conducive to disease severity, and other variables in their geographical analyses.”

Example 2

Lines 159 - 163 -

"The field survey was carried out by means of a systematic line survey combined with the investigation of standard plots. The area with more than 10% affected trees was found to set up the standard plots, and 0.2 hm2 standard sites were set up every 5 hm2. Affected trees with the representative symptomatic were selected as samples to further isolate and culture pathogenic bacteria in the standard field."

Possible way to improve the English and better communicate what was actually done to collect the data is as follows. 

“The field survey was carried out using a systematic line survey and augmented with data collected from standard circular(?) plots (0.2 hm2). The additional plots were established when a surveyed area was found to have more than 10% affected trees. Plots were set up every 5 hm2 for these situations. Representative trees with poplar canker were selected for sampling and verification of the causal fungus. Specifically, isolations were made from stem cankers in the laboratory to confirm presence of C. chrysoperma.”
